# Bugs in Bugs: The Role of Probiotics and Prebiotics in Maintenance of Health in Mass-Reared Insects

**DOI:** 10.3390/insects13040376

**Published:** 2022-04-11

**Authors:** Carlotta Savio, Loretta Mugo-Kamiri, Jennifer K. Upfold

**Affiliations:** 1University of Paris Saclay, INRAE, Micalis, GME, 78350 Jouy en Josas, France; jennifer.upfold@inrae.fr; 2Laboratory of Entomology, Wageningen University, 6708 PB Wageningen, The Netherlands; 3Institut de Recherche sur la Biologie de l’Insecte, UMR 7261, CNRS-University of Tours, 37200 Tours, France; lorettawangui.mugo@etu.univ-tours.fr; 4Centre for Ecology and Conservation, Penryn Campus, College of Life and Environmental Science, University of Exeter, Cornwall TR10 9FE, UK; 5Department of Plant and Environmental Science, University of Copenhagen, Thorvaildsensvej 40, 1871 Frederiksberg, Denmark

**Keywords:** probiotics, prebiotics, mass-reared insects, insect diseases, microbiota, performance, health

## Abstract

**Simple Summary:**

The importance of insect farming is increasing in the livestock market by evolving into a form of intensive production that is often characterized by a high density of individuals kept in closed environments. These conditions can cause a higher risk of occurrence of insect diseases and the lowering of reproductive and growth performances. The role of microbiota composition in insect behaviour and health maintenance could be further studied for selecting microorganisms that act as probiotics for the main mass reared insect species. These probiotics could enhance host performances and reduce the incidence of risks related to insect diseases.

**Abstract:**

Interactions between insects and their microbiota affect insect behaviour and evolution. When specific microorganisms are provided as a dietary supplement, insect reproduction, food conversion and growth are enhanced and health is improved in cases of nutritional deficiency or pathogen infection. The purpose of this review is to provide an overview of insect–microbiota interactions, to review the role of probiotics, their general use in insects reared for food and feed, and their interactions with the host microbiota. We review how bacterial strains have been selected for insect species reared for food and feed and discuss methods used to isolate and measure the effectiveness of a probiotic. We outline future perspectives on probiotic applications in mass-reared insects.

## 1. Introduction

Insect mass-rearing for food and feed has been identified as a valuable industry with current predictions expecting the market to grow by 47% from 2019 to 2026 [1]. This growth will see production volumes reaching an expected 730,000 tonnes by 2030 [2]. Over 1900 insect species have been reported in literature as being consumed worldwide [3]; however, only a few of these species are mass-reared in a more intensive manner. Much like traditional intensive livestock production, mass-rearing of insects faces similar challenges of high densities, a high rate of pathogen transmission, higher susceptibility to pathogens due to lack of oxygen, high temperatures, and nutrient deficiencies [4,5]. These challenges associated with farming insects need to be investigated with urgency in order to develop a successful industry, which has the potential to contribute to global food security.

A solution to mitigate some of these challenges has been the administration of antimicrobials, but this comes with the potential risk of emerging multiple antibiotic resistant ‘superbugs’ that threaten both animal and human health. The trend has therefore shifted to the use of probiotics which are defined as “live microorganisms that, when administered in adequate amounts confer a health benefit on the host” [6]. It is now apparent that every organism is associated with a microbial community ranging from parasitic to mutualistic, and this is true for all animals, from humans to invertebrates, insects included. In particular, the gut microbiota has been of focus to researchers in recent years due to its link to the health status of its host, with more and more studies finding that exploiting these microorganisms may improve animal productivity and maintain their health and wellbeing [7,8].

Interactions between insects and their microbiota play an important role in behaviour and evolution of many insect species [9,10]. Several microorganisms are able to manipulate host behaviour to increase their transmission. For example, *Wolbachia* which is able to modify the mating preference of its hosts when it acts as a symbiont, and the lack of microbiota or the presence of foreign gut bacteria can distort the feeding behaviour of insects by changing their sense of smell [9]. Reproduction, conversion, and growth performances have been related to specific microorganisms in mass-reared insects [11]. Insect–microorganism communication is bi-directional and social interactions in insects can impact the distribution of microorganisms within the population.

Insect diets have, in this context, an important role in providing nutrients both to the insects and the microorganisms. Within its composition, it is possible to highlight specific nutrients that act as prebiotics and that have been defined as “selectively fermented ingredients that allows specific changes, both in the composition and/or activity in the gastrointestinal microbiota that confers benefits upon host wellbeing and health” [12].

Our goal is to provide an overview of the use of probiotics, and in brief, prebiotics, in the rapidly growing industry of insects for food and feed. As previously mentioned, there are over 1900 species of insects consumed worldwide [3]; however, along with some other relevant examples, this review will focus on the important species currently mass reared in this industry, which includes *Acheta domesticus* (L.) (Orthoptera: Grylloidea), *Gryllus bimaculatus* (De Geer) (Orthoptera: Gryllidae), *Hermetia illucens* (L.) (Diptera: Stratiomyidae), *Musca domestica* (L.) (Diptera: Muscidae), *Tenebrio molitor* (L.) (Coleoptera: Tenebrionidae), as well as two other insects of economic importance that are mass-reared, *Bombyx mori* (L.) (Lepidoptera: Bombycidae) for their role in sericulture, and *Galleria mellonella* (L.) (Lepidoptera: Pyralidae), for their role as a non-mammalian model for the study of human pathogens.

This review will highlight the relationship between insects and their gut microbiome and discuss the mechanisms by which probiotics may exert their beneficial effects. It also reviews some of the methods that have been used to reduce the occurrence of disease in reared colonies and gives a summary of the probiotics that have been tested in the seven insect species mentioned. Lastly, we highlight some of the techniques used in isolation of probiotics and ways of testing microorganisms for probiotic potential as well as future perspectives on industrial applications.

## 2. Defining Probiotics and Prebiotics: The Bugs That Debug the Bugs

Every organism is associated with a microbial community that may promote its health. The identification of probiotics opened the possibility of exploring the health of insects when they are provided with beneficial microorganisms. In particular, the effect of the microorganisms is assessed for improving growth and reproductive performance and for decreasing the occurrence of diseases in stressful rearing conditions. The World Health Organization’s internationally endorsed definition of probiotics is “live microorganisms that, when administered in adequate amounts confer a health benefit on the host” [6]. This definition, however, is still unclear in some circumstances, thus causing controversy and confusion. A distinction is made between microorganisms given to insects as a supplement and those that are commensal gut microbes that putatively confer health benefits to the insect [13]. The latter are often erroneously termed as probiotics, but this requires that they be isolated and characterized and their subsequent health-promoting effects validated [13]. Other terms that have been used synonymously are direct-fed microbials given in animal diets and live biotherapeutic products that are more pharmaceutical and take the form of drugs rather than food supplements, even though they are intended for the same use. Overall, the scope of probiotic intervention is expanding, leading to various ways in which the product reaches the market and different regulatory requirements, and this comes with various terms/definitions as a result.

The concept of prebiotics came to light when, in the early 1950s, scientists discovered that there was a special growth-promoting factor in human milk that aided the growth of the probiotic *Bifidobacterium bifidus* (Tissier) [14]. These components were later named by Gibson and Robertfroid [15] as prebiotics and defined as “nondigestible food ingredients that beneficially affect the host by selectively stimulating the growth and/or activity of one or a limited number of bacterial species already resident in the colon, and thus attempt to improve host health”. To put it another way, these are the nutrients that feed the probiotic microorganisms enabling their proliferation in the gut [16]. These nutrients are mostly fibers, and they include inulin, oligofructose (produced from inulin), and fructooligosaccharides (FOS) synthetically produced from sucrose, as well as galactose-containing and xylose-containing oligosaccharides, resistant starch (RS), pectin, and other fermentable fibers [17]. It was not long before probiotics and prebiotics were conveniently combined in one synergistic pack known as synbiotics [15,18]. For the insect mass rearing industry, this highlights the fundamental importance of different types of diet and diet quality for the success of the probiotic application.

## 3. The Crosstalk between the Insect and Their Intestinal Microbiota

The ubiquitous nature of gut bacteria and increasing knowledge of their numerous advantages to insect hosts has led to their application as probiotics in the insect mass rearing industry. The use of probiotics is based on the interaction between the host and their gut microbes; hence, understanding the nature of this crosstalk is fundamental to understanding the process. The insect gut microbiota and their collective genomes (‘microbiome’) have captured the interest of many researchers today, as an ‘organ’ in itself that plays a core role in influencing key insect traits [19]. The insect microbiome consists of a large diversity of microorganisms including bacteria (bacteriome), fungi (mycobiome), viruses (virome), and archaea (archaeome), but bacteria are the most abundant and the most studied [19].

Studies examining this spectrum of symbiotic relationships have pointed to the beneficial effect of a variety of bacteria and yeast species in different insects and thus their application as probiotics. For example, probiotic application of *Klebsiella oxytoca* and an *Enterobacter* strain increased larval growth of *Ceratitis capitata* (Wiedemann) (Diptera: Tephrtidae) used in Sterile Insect Technique (SIT) application [20] by increasing larval growth, especially due to bacterial synthesis of nutrients and protection against pathogens through the release of some antimicrobial compounds. A similar effect was also observed in transgenic *Plutella xylostella* (L.) (Lepidoptera: Plutellidae) where aseptically reared larvae that otherwise had low pupal weight and poor survival registered a substantial increase in pupal weight and male fitness when inoculated with *Enterobacter cloacae* (Jordan) [21]. However, to initiate any form of host–microbe interaction, a series of steps occur beginning with the microbiota acquisition followed by gut colonization or adhesion and progressing to an establishment in the gut and further transmission back to the environment or to new hosts [22]. It has been shown that insects can acquire their microbiota horizontally from the environment mainly through diet [23,24]. Dietary habits have been shown to affect both the composition and robustness of gut communities by regulating nutrient availability for the microbes [25]. Microbiota are also acquired by social interaction through trophallaxis and coprophagy transplantation as seen in termites and bees, and thirdly, through vertical transmission from parent to offspring via the egg surface, which is exposed to microbes from the ovaries of the mothers [26].

Adherence of the microorganisms to the gut lumen then follows, but the mechanisms vary across the different species, likely due to diversity in the physiology, morphology, and ecology of insects [27,28]. Gut colonization is affected by many different factors, including physicochemical gut conditions, i.e., pH, redox potential, and oxygen content. Gut compartmentalization can affect microbiota distribution presenting increased microbial density from anterior to posterior compartments. Moreover, the presence of enzymes and immunological compounds in the gut and the life history characteristics causes changes in community abundance [29,30]. Depending on the insect species, different bacterial communities have to develop strategies that allow them to survive and persist in the harsh conditions of the host such as the highly alkaline guts of lepidopteran species. To illustrate this, an RNA-sequencing study showed that the gut symbiont *E. mundtii* had upregulated pathways for tolerating high alkaline stress during its passage in the gut of *Spodoptera littoralis* (Boisduval) (Lepidoptera: Noctuidae) [31,32].

After successful colonization, these microbes may participate in many different symbiotic, pathogenic, or vectoring activities within the host [7,33]. Focusing mainly on the symbiotic or mutualistic roles, microbes play a key role in metabolism by breaking down indigestible plant-derived polysaccharides through microbiota encoded carbohydrate-degrading enzymes in the midgut and hindgut of caterpillars (Lepidoptera), termites (Dictyoptera: Isoptera), honey bees (Hymenoptera), beetles (Coleoptera), crickets (Orthoptera), and other herbivorous insects [34]. These enzymes include cellulases, hemicellulases, and pectinases that are generally absent in insects [35]. Remarkably, recent phylogenetic studies have shown that the genes encoding these enzymes in gut microbes have also been encoded in the genomes of some hosts such as the mustard leaf beetle, *Phaedon cochleariae* (Fabricius) (Coleoptera: Chrysomelidae), signifying horizontal gene transfer [36,37]. The breakdown of these plant cell wall materials culminates in the formation of short-chain fatty acids that impact both the microbe and host’s nutrition and help in maintaining the integrity of the gut barrier [27,34]. In this way, gut microbes also compensate for the sometimes nutrient-poor diets of their hosts through nutrient provisioning, for instance, *Buchnera aphidicola* (Munson et al.) in pea aphids that provide essential amino acids that are lacking in the insect diet [38].

Through provisioning of nutrients and aiding digestion, gut symbionts then positively influence the growth and development of hosts, as shown in numerous studies. A study in *Drosophila melanogaster* (Meigen) (Diptera; Drosophilidae), for example, showed that aseptically reared insects had reduced growth and slower development as compared to conventionally reared insects. However, when inoculated with *Acetobacter pomorum* (Sokkollek), a gut commensal, the growth and development was restored to a similar rate as that of conventionally reared insects [19]. It was also observed that the addition of *Leuconostoc* spp. as a probiotic in the diet of the fruit fly *Bactrocera tryoni* (Froggatt) reduces the mean development time from egg to adult [39].

During herbivory, insects also encounter a variety of toxic plant defense chemicals, the detoxification of which is aided by their gut symbionts which enables their success as pests [40]. This type of detoxifying-symbiosis confers resistance not only to plant allelochemicals but also to insecticides [41]. This is achieved through diverse mechanisms, for example, enzymatic degradation of potential toxic phytochemical compounds as aglycones by gut symbionts of *T. molitor* [42,43]. Gut symbionts also play a protective role by increasing the host’s resistance to pathogens. Several mechanisms come into play here, including the inhibition of infection, as seen in Tsetse fly *Glossina morsitans* Westwood (Diptera: Glossinidae) colonized with the commensal *Kosakonia cowanii Zambiae* which inhibits *Serratia marcescens* and *Trypanosoma* by raising the pH of the gut [26]. Gut symbionts also protect the host by nutrient and space competition with invading pathogens, thus edging them out as observed in mammals and also suspected to be the case in insect hosts [34]. The microbiota also enhances the epithelial barrier function to prevent systemic infection by pathogens, as seen in mosquitoes [44] and the stimulation of the host immune system or immune priming [34].

Overall, the host–microbiota relationship is intricate and intriguing but there are still some aspects that are yet to be explored and well understood for instance the influence of abiotic factors on the interaction and how microbiota impact insect population dynamics as well as behaviour manipulation, all of which could be advantageous in the advancement of probiotic use in the insect mass rearing industry.

### Probiotics Applications as a Means of Decreasing Disease Occurrence in Mass Reared Insects

Mass-reared insects are highly susceptible to diseases caused by organisms that belong to different Phyla within bacteria, viruses, fungi, protista, and nematoda [45]. Finding a method for preventing these diseases and increasing growth and reproductive performances in insects reared for food and feed purposes is thus a key goal of both industries and researchers [46]. Maciel-Vergara et al. [5] presented an overview of good production practices for reducing risks of pathogen occurrences such as daily adherence to good hygiene practices, differential breeding, mechanical pest control, and techniques such as heat shock/thermal therapy, breeding of tolerant strains, biological control, and RNA interference. In this context, probiotics have also been considered an option for decreasing the impact of diseases due to their ability to positively influence host performance and enhance immune responses against pathogens [5,47].

The most known probiotic bacteria are Lactic Acid Bacteria (LAB) that present a high immune system activity in humans [48]. In insects, antibacterial activity and immune regulatory effects have been widely recorded within the *Lactobacillus* genus in the microbiota of silkworm *Bombyx mori* and the honeybee *Apis mellifera* (L.) (Hymenoptera: Apidae) when infected with the pathogen *Pseudomonas aeruginosa* (Gessard) and *Nosema* spp., respectively [49,50]. Observations of *Galleria mellonella* have highlighted the antimicrobial effects of *Lactobacillus reuteri* and *Lactobacillus rhamnosus* on *Staphylococcus aureus*, *Escherichia coli*, *Candida albicans* and *Pneumonia aeruginosa* [51,52,53]. Probiotics strains have been tested against *Nosema ceranae* on honeybee alongside small-molecule RNA interference techniques and supplements for decreasing the bee spore load and viability and increasing its survival and performance [54]. At the genus level, *Lactobacillus* and *Bifidobacteria* had an impact on the pathogens, but when *L. rhamnosus* and sucrose were provided to honeybees to decrease the impact of *N. ceranae*, higher mortality rates and lower phenol oxidase production were recorded [54,55,56]. The positive effects of *Lactobacillus* spp. on hosts infected by fungi have been highlighted in studies focused on *Drosophila melanogaster* and *Galleria mellonella*, respectively, infected with diaporthe FY and *C. albicans* [8,57]. Other genera such as *Enterococcus* have been studied for decreasing the occurrence of bacterial diseases. *E. mundtii* showed positive effects on *Tribolium castaneum* (Herbst) (Coleoptera: Tenebrionidae) immune responses against *Bacillus thuringiensis* infections [58]. As for viral diseases, the activation of insect-specific or generic immune responses by probiotics is still to be clarified [59]. Although endosymbionts do not fall under the definition of probiotics, it is worth mentioning that the main positive results are related to the presence of *Wolbachia* and *Spiroplasma* that increase resistance to viral diseases in *T. molitor* and *G. mellonella* [59,60]. It is interesting to note, however, that some bacteria species isolated from shrimps have been shown to have antiviral activity in a plaque assay and demonstrated positive effects on fitness performance in shrimps infected with white spot syndrome virus, which sheds light on some possible techniques that could be used in screening insect microbiota for the same [16,61].

Although the use of probiotics in disease management is very promising, it is still hampered by a scarcity of knowledge on insect-pathogen dynamics and the influence of the stressors on production performances and on insects’ susceptibility to diseases. This calls for the need to maintain a holistic approach in the general management of mass rearing systems, taking into account environmental factors, diet, microbiota and genetic factors.

## 4. The Most Common Microorganisms Used as Probiotics in Insects

The most commonly tested probiotics for insects reared for food and feed belong to the genera *Lactobacillus*, *Saccharomyces*, *Streptococcus*, and *Bacillus* which, according to the World Gastroenterology Organization [48], are among the seven ‘core’ microorganisms most often used as probiotics. *Enterococcus* is a largely underrepresented group, especially given that it is a common symbiont of the insect gut and especially in Lepidoptera. This may be in part due to the fact that *Enterococcus* species have both pathogenic and probiotic strains [62,63]. Nevertheless, an example of its potential use as a probiotic was described by Grau et al. [58] when they isolated an *E. mundtii* strain from the feces of *Ephestia kuehniella* (Zeller) (Lepidoptera: Pyralidae), which showed antimicrobial activity against a variety of Gram-positive and Gram-negative bacteria, and increased survival in *Tribolium castaneum* beetles after infection with *Bacillus thuringiensis*.

Aside from summarizing probiotics tested for insects reared for food and feed, Table 1 highlights the lack of research into probiotics for crickets. However, the primary disease observed in reared *A. domesticus* populations is caused by a densovirus (AdDNV). An effective approach to reduce the impacts of the virus on *A. domesticus* populations may not be through the use of probiotics but rather through RNA interference technologies. This approach was used by La Fauce and Owens [64] on *A. domesticus* to reduce PmergDNV titres and subsequent mortality from the virus, by feeding the insect with dsRNA specific to the capsid protein by mixing it into their food. Similarly, there has been little work into probiotics for *Acheta domesticus* (L.) (Orthoptera: Gryllidae), another cricket species mass reared in particular for pet food and human consumption. However, more towards the direction of prebiotics, a recent study found the incorporation of Jew weeds, *Comellina sinensis* (L.) Kuntze, into the diet of the cricket usually just fed chicken feed, resulted in an increase in body weight and improved microbial quality [65].

### 4.1. Isolating Potential Probiotic Strains and Their Characterization

There is no standard method for identifying probiotic strains; however, potential probiotic candidates are usually identified in the core microbiota of the insect [84,85,86], primarily through a metagenomic approach, as illustrated in Figure 1 The microbiota is readily characterized via DNA extraction followed by 16S rRNA gene sequencing. The 16S rRNA gene region is used for sequencing, as it is a short (approximately 300–500 bases), conserved gene specific to bacterial genus, and even some species [87]. However, as discussed in a recent review by Winand et al. [88], next-generation sequencing such as Illumina and Nanopore Technologies offers a reliable identification of bacterial genera but can have reduced accuracy in the identification of bacterial species, necessitating the combination of omics with classical microbiological techniques to get down to species and strain level. Once the core microbiota is classified, further culturing steps can be utilized to target specific bacterial isolates. Yeruva et al. [89] utilized this approach when they assessed the midgut of *B. mori* to identify potential probiotics. Through this approach, *Enterococcus*, *Lactobacillus*, and *Bacillus* species were found to be dominant in the microbiota, and upon further evaluation, these species are well-known producers of coenzymes, antimicrobial substances and extracellular enzymes [89].

The analysis of health-promoting properties of probiotic bacteria and yeast should be conducted using in vitro and in vivo approaches before providing the strain on a large scale [90]. Papadimitriou et al. [91] and Byakika et al. [92] provided an overview on the assays that can be performed on bacterial strains for observing if the safety and biological and chemical characteristics of the strain fulfill the characteristics of being classified as probiotics. The same procedures have to be performed on yeast strains [93].

#### 4.1.1. Safety Assays

All intrinsic characteristics relevant to the strains have to be fully evaluated before considering the strain safe for probiotic purposes. Safety assays on the production of biogenic amines by the decarboxylation of amino acids by substrate specific decarboxylases of potential probiotic bacteria should be conducted; for example, the production of histamine, which can persist in the food chain and lead to severe allergies through consumption of edible insects [91,94].

Moreover, the determination of the minimal inhibition concentrations (MIC) related to antimicrobials in probiotic strains reduces the risk of the addition of antimicrobial-resistant genes in the insects’ gut environment and avoidance of the horizontal transmission of these genes to other microorganisms. The official protocols created by the European Food Safety Agency [95] have to be performed following international standard recognized methods cultivating the strain on seven antibiotics and the cut-off values, taking into considerations strain, growing conditions, and dilution variability (CLSI; www.clsi.org accessed on 2 February 2022; ISO; www.iso.org accessed on 15 January 2022). Other safety considerations include the production of virulent genes and toxin production, which could be deleterious to reared insects and consumers of insects.

#### 4.1.2. Analysis of Antimicrobial Potential

The analysis of the antimicrobial potential is performed to assess the strain’s ability to produce antimicrobial compounds (AMCs) active against selected pathogens and can be performed by running in vitro assays. The agar spot method is efficient for recording the probiotic’s zone of pathogens’ inhibition on selected media in selected growth conditions [53,78,95]. The technique is used for quantifying the effect of antimicrobial agents such as bacteriocins and organic acids on selected pathogens. As agar gradient and culturing conditions can distort the effective concentration/production of the molecules, other methods can be applied. The paper-disk diffusion assay [96] and the well diffusion assay work on the same principle. For reducing the effect of the concentration of agar, liquid-medium techniques have been proposed by recording nisin production in *Leuconostoc mesenteroides* (De Moss et al.) [97,98]. Microbiota interactions and environmental conditions such as pH and temperature can influence the production of antimicrobial compounds. For instance, in vitro studies highlighted optimal conditions of pH 6.2 and a temperature of 37 °C for some *Lactobacillus* spp. for producing bacteriocins [99]. It is also important to take into consideration the consistent differences that are present within and between insect orders in terms of gut structure and gut environment, which also affect the production of AMCs [19].

As probiotics can decrease pathogen inference by competition, co-aggregation abilities can be recorded by in vitro assays based on absorbance measurements and fluorescence and radiolabelling detection [100]. Other in vitro methods used for observing the microbiota interactions and competition for space acquisition are focused on autoaggregation and on the probiotic’s effects on the pathogen’s capacity to produce biofilms. The methods are based on spectroscopy measurements. The autoaggregation of *Lactobacillus* spp. varies from 10 to 23% influencing competition processes against pathogens, therefore decreasing the ability of microorganisms such as *L. monocytogenes* to produce biofilm and to infect the host [92,101,102]. Hydrophobicity properties related to gut adhesion and colonization can be measured by performing spectrometry on isolated strains of the potential probiotic culture in several organic solvents [92,102,103]. Another important parameter that can influence gut colonization by adhesion is the production of exopolysaccharides (EPS). For in vitro analysis of the production of these molecules, an extraction followed by concentration estimation by phenol-sulfuric acid method is efficient [102]. The expression of these parameters increases the probiotic strain capabilities to compete with pathogens strains in gut space colonization by reducing the chance of incurring infection.

#### 4.1.3. Assessing Immune Modulation

The influence of a potential probiotic strain on the immune response can be determined by monitoring the expression of important immune response genes, or by monitoring the expression of genes encoding immunologically important molecules by quantitative RT-qPCR [84]. The innate immune system comprises a set of genes representing four immune system pathways (Toll, Imd, JNK and JAK/STAT) [104]. The most commonly investigated groups of genes for probiotic studies are antimicrobial peptides (AMP) and pattern-recognition receptors of the Toll and Imd pathways due to the receptors mediating host-microbiota communication [86]. Coupled with an assessment of immune-relevant gene expression, it is also common practice to collect hemolymph samples for measuring other immunomodulatory parameters. Ordinarily, these are assays involving measuring the level of phenoloxidase, total protein concentration, and total hemocyte counts and differential haemocytes circulating in the hemolymph after the administration of a probiotic [81,105].

### 4.2. Ecological Fitness Assay

According to Peacock [106] fitness can be most usefully understood as all aspects resulting in survival, not only the properties of reproductive success. Furthermore, Rosenberg and Bouchard [107] clarify ecological fitness as interactions between organisms and environments. With this definition in mind, it is clear that the experimental design is critical when evaluating the effect of probiotics on overall fitness and health, as the results can be influenced by several factors: 1. method of delivery; 2. biological traits measured; 3. effect on different life stages; 4. pre-existing microbiota; and 5. diet [108]. Additionally, due to the diversity of life-history strategies and environments in which insects inhabit, there can be species-specific influences within these factors, further establishing a criterion for assessing probiotics in insects complicated. Nevertheless, indicators of biological fitness are often measured by the overall longevity, mortality, fertility, and fecundity of the insect. For those insect species that are holometabolous, the weight of pupae and adult emergence rate can also be monitored, as can, in some cases, the flight capacity [102].

Within these measurements, it can be important to distinguish the additive role of the probiotic on having an effect due to the interaction with the insect or simply as a source of nutrients. Some researchers have attempted to separate these responses by providing both dead and live probiotics. When assessing the effect of *Klebsiella oxytoca* and *Enterobacter* sp. AA26 as probiotics in larval and adult *C. capitata* mass-reared for SIT, Kyritsis et al. [109] provided both dead (inactivated via heat treatment) and live bacteria. In doing this, they were able to differentiate the responses as an effect of the live bacteria, or not. A reduction in the developmental time of the immature stages was found for flies fed both dead and live *K. oxytica*-enriched diets, but a positive effect on flight ability was only demonstrated in individuals provided the live bacteria. Similarly, Gavriel et al. [110] only found an increase in mating competitiveness in medfly adults provided live bacteria, with no beneficial effect on males fed dead bacteria. In general, this highlights the need to better understand the role of a supplement as either providing an additional source of nutrients, or as establishing in the gut and interacting with the host.

In summary, there is no ‘best practice’ for assessing the efficacy of probiotics in insects. However, assays involving the assessment of the effect of the probiotic on biological parameters have become a standard practice. An often-overlooked element in categorizing probiotic potential is the evaluation of the microbiota community composition after the administration of the probiotic as some strains are able to reset unbalanced microbiota phenotypes by modulating the host defense systems, as well as drive microbiome functional shifts. These shifts, if sufficient, could achieve a positive effect or trigger a decline in nutrients, energy, and metabolic activity of the insect and reduce overall growth and reproduction performance [85,111].

## 5. Improving Mass Reared Insect Fitness by Probiotic Provision

The gut microbial composition has the potential to shape its host growth trajectory in a stressful environment. The manipulation of the gut microbial composition by providing selected probiotic feed additives in the rearing systems could be a way to improve insect fitness, reduce the effects of external factors such as stress, and reduce or altogether prevent the use of chemical growth promoters [112]. Studies on aseptically reared insects have mainly focused on the role of specific strains on nutrient absorption in poor nutrient conditions. A study on the popular insect model *Drosophila melanogaster* highlighted the importance of the beneficial metabolic dialogue between *Lactobacillus plantarum* and *Acetobacter pomorum*. The provision of these bacteria determines the boosting of the host juvenile growth despite the malnutrition, by each providing essential metabolites such as lactate, essential amino acids, and anabolic metabolites that foster growth [113].

Even if the selection of specific strains for improving fitness performances is still ongoing for several mass-reared insect species, some results have already been obtained on *B. mori*’s body weight, cocoon, shell, and pupation rate with the addition of *Lactobacillus* species in the diet [89]. *Saccharomyces cerevisiae*, *Staphylococcus gallinarum* and *Staphylococcus arlettae* provided on mulberry leaves resulted in better performance in *B. mori* [70,71]. Positive connections between the provision of the strain *Pediococcus pentosaceus* (Figure 2) to *T. molitor* larvae and fitness performances have been proven allowing its definition as probiotic [78]. The definition of a protocol for providing the strain on an industrial scale is ongoing. Initial studies on *T. molitor* have demonstrated that the provision of a mixed culture of probiotic bacteria can affect growth and weight gain positively [114].

The primary interests of providing probiotics to *Hermetia illucens* are mainly focused on waste conversion and their positive effect on larval growth. *Arthrobacter* AK19 and *Rhodococcus rhodochrous* 21198 increased the protein digestion and absorption by 20–30% with no impact on the microbial community. On the other hand, the provision of *Bifidobacterium breve*, caused an increase of 50% of larval final weight, 20% lower waste conversion, and the suppression of microbial community diversity at a benchtop and industrial scale [79].

The nutritional content of the insects is affected by the manipulation of the microbial composition. The dry matter and crude protein percentage showed higher values in *T. molitor* larvae and *H. illucens*’s fatty acids compositions and presented a shift to polyunsaturated fatty acids [79,114]. The selection of targeted microorganisms plays a key role in shaping the microbial community and obtaining positive effects on fitness parameters.

## 6. Concluding Remarks and Future Perspectives

The occurrence of diseases in mass-reared insects and the need of reducing the use of antibiotics in all the production systems to cope with the rising frequency of antibiotic resistance, are demanding new solutions for preserving animals’ health and improving their fitness performances. The identification and the selection of host-specific probiotics could represent a sustainable solution for stabilizing insect production and ensuring food and feed safety.

Once the probiotic strain is selected, the supplements’ formulation and the role of prebiotics in synbiotic interactions could have an important role in stabilizing the commercial product and in assuring the probiotic gut colonization and persistence after the provision to the insects.

Insect–microbiota relationships can affect behaviour and fitness performances in several stressful situations. Interest in the probiotic provision of insect species for food and feed purposes has already led to the enhancement of fitness performances and immune responses. Therefore, the chance to shape the microbial community favouring probiotic microorganisms by providing prebiotics in the diet increase the opportunities to promote the health status of the insects and to decrease the occurrence of diseases in mass reared conditions. A lot of questions related to synbiotic–host relationship are still open and multivariate statistical models are needed for studying the effects of diet, environmental factors, and microbiota on these interactions. Further studies could be focused on the manipulation of mass-reared insect microbiota for breeding individuals with a selected starting gut microbiota that could allow better growth and reproductive performance, decreasing the occurrence of diseases.

## Figures and Tables

**Figure 1 insects-13-00376-f001:**
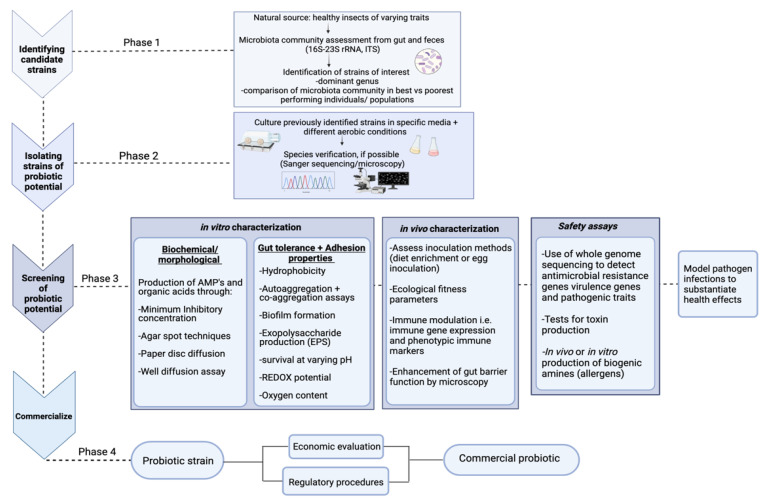
General workflow and screening techniques to characterize strains probiotic potential prior to commercialization. Phases 1 and 2 display the identification of candidate strains and isolation methods; phase 3 offers different in vitro and in vivo techniques for characterizing the probiotic potential. Fundamental in vitro safety assays are also listed. Phase 4 highlights two major factors that also need evaluation prior to the probiotic being used commercially.

**Figure 2 insects-13-00376-f002:**
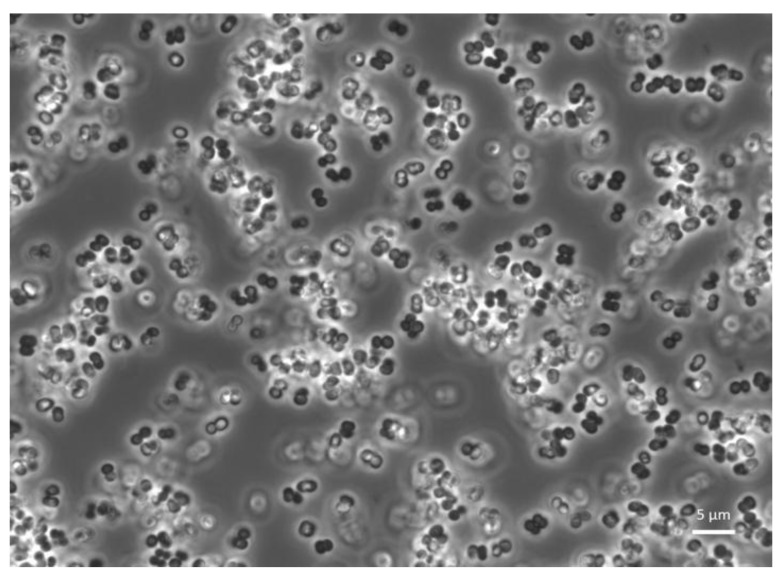
Confocal microscopy observation of *Pediococcus pentosaceus* KVBL 19-01. Probiotic activity of the strain has been recorded on *T. molitor* by Lecocq et al. [78].

**Table 1 insects-13-00376-t001:** A summary of probiotics tested on insects mass-reared for food and feed with the objective of improving insect performance or fitness against natural pathogens in the mass rearing environment. The table does not include data where bacteria/yeast have been provided to the insect as a probiotic to test its efficacy against a specific human pathogen in vivo, nor does it include insects reared for sterile insect technique programmes.

Insect Species	Probiotics	Effects on Performance and Yield	Ref.
Silkworm	Bacteria		
*Bombyx mori*	** *Bifidobacteria* **		
	*Bifidobacterium bifidum*	Found to be an immunomodulating agent (increase in the activity of protease, amylase and invertase); increased raw silk production with fewer cocoons	[66]
	** *Lactobacilli* **		
	*Lactobacillus acidophilus*	Stimulated growth factors leading to an increase in the silk yield and to an improvement of the silk harvest	[67]
	*L. casei*	Improved larval weight, cocooning ratio, pupation ratio, and economic characters (cocoon weight and size) when larvae were infected with microsporidium *Nosema bombycis*	[68]
	*L. plantarum*	Helped to increase body weight, cocoon, shell, and pupation rate	[69]
	** *Staphylococci* **		
	*Staphylococcus gallinarum strain SWGB 7* & *S. arlettae strain SWGB 16*	Increased larval growth and cocoon characters (filament length and weight, finer denier)	[70]
	** Yeast **		
	*Saccharomyces cerevisiae*	Immunomodulating agent; increased raw silk production with fewer cocoons; increased protein content	[66,71]
	** Fungi **		
	*Trichoderma harzianumas*	Improved food digestion leading to increased growth and resistance to mortality by *Metarhizium anisopliae* and *Beauveria bassiana*	[72]
	** Commercial products **		
	*Lact-Act ^a^*	Larvae reared on leaves sprayed with Lact-Act had increased survival when exposed to bacterial pathogens (*Bacillus thuringiensis* var. *sotto.* and *Staphylococcus aureus*)	[73]
**Insect species**	**Probiotics**	**Effects on performance and yield**	**Ref.**
**Greater** **wax moth**	** Bacteria **		
*Galleria mellonella*	** *Clostridiaceae* **		
	*Clostridium butyricum* Miyairi *588*	Induced immune response and increased survival rates against *Salmonella enterica* serovar *Typhimurium,* enteropathogenic *Escherichia coli* or *Listeria monocytogenes.*	[74]
	** *Lactobacilli* **		
	Lactobacillus acidophilus ATCC 4356	Increased survival from *Candida albicans* infection	[75]
	*L. kunkeei ^b^*	Reduces infection of *Pseudomonas aeruginosa* through biofilm formation and affecting their stability	[76]
	*L. rhamnosus* ATCC 7469	Promoted greater protection in larvae infected with *Staphylococcus aureus* or *Escherichia coli*.	[52]
	*L. rhamnosus* ATCC 9595	Reduces infection of *Pseudomonas aeruginosa* through biofilm formation and affecting their stability	[51]
	*L. rhamnosus* GG	Induced immune response and increased survival rates against *Salmonella enterica* serovar *Typhimurium,* enteropathogenic *Escherichia coli* or *Listeria monocytogenes.*	[74]
**Yellow mealworm**	** Bacteria **		
*Tenebrio* *molitor*	** *Bacilli* **		
	*Bacillus subtilis*	Enhanced growth and nutritional fortification	[77]
	*B. toyonensis*	Enhanced growth and increased dry matter weight of produced feed	[77]
	** *Enterococcaceae* **		
	*Enterococcus faecalis*	Increased larval weight gain and overall size and shorter time to pupation, also increased the crude protein content	[77]
	** *Lactobacilli* **		
	*Pediococcus pentosaceus* (Isolated from the gut of *Tenebrio* larvae)	Reduces mortality in larvae and accelerates the rate of development. The strain has antimicrobial activity towards a number of pathogenic bacteria including several *Bacillus thuringiensis, Serratia,* and *Pseudomonas spp*.	[78]
**Insect species**	**Probiotics**	**Effects on performance and yield**	**Ref.**
**Black** **soldier fly**	** Bacteria **		
*Hermetia* *illucens*	** Actinomycetia **		
	*Arthrobacter AK19*	Enhanced growth rate at early life stages culminating in larger larvae than control	[79]
	** *Bacilli* **		
	*Bacillus subtilis S15 S16 S19;* *B. subtilis natto D1*	Increased larval weight and total development time compared to control larvae	[80]
	** *Bifidobacteria* **		
	*Bifidobacterium breve*	Larvae had lower weights and appeared weak/slow/discolored compared to control	[79]
	** *Nocardiaceae* **		
	*Rhodococcus rhodochrous*	Increased conversion rate, which could result in larger larvae with less feed. Larvae had increased proteins content related to energy production and storage. Larvae without the probiotic which had higher content of proteins related to stress responses.	[81]
	** Commercial product **		
	Actisaf^®^ Sc47 *^c^*	Increased bioconversion rate, lipid and protein yield in processed larvae	[82]
**House fly**	** Bacteria **		
*Musca* *domestica*	** *Enterobacteriaceae* **		
	*Enterobacter hormaechei*	Increased body length and weight, pupal weight, and shortened growth cycle, which is a considerable advantage that can contribute to cost savings and boost production in large-scale feeding facilities.	[83]

*^a^*—Probiotic powder containing *Lactobacillus sporogens*, *Bacillus thuringiensis*, yeast hydrolysate, a-amylase, vita. min and mineral mix; *^b^*—Strain was isolated from honeybee guts and tested against gram—pathogen *Pseudomonas aeruginosa*; *^c^*—Yeast—*Saccharomyces cerevisiae* CNCM I-4407.

## Data Availability

Not applicable.

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
