# Peer review of "Bugs in Bugs: The Role of Probiotics and Prebiotics in Maintenance of Health in Mass-Reared Insects"

_insects, 2022, doi:10.3390/insects13040376_

Round 1

Reviewer 1 Report

The manuscript reviews the current state of knowledge on probiotics for insects that are mass-reared for food and feed. It offers valuable recommendations for the future development of this important aspect of mass-rearing that will help to reduce pathological effects of some microorganisms and toxins that can affect insect mass-rearing. The review is timely and will likely shape the way that this work continues in the future.

I have relatively minor comments that should be addressed:

1. One way in which insects can benefit from the gut microbiota is through detoxification of insecticides. It would be worth mentioning this when discussing plant secondary metabolite detoxification in section 3 because insects reared for food and feed on plant and vegetable waste may come into contact with insecticide residues. Having a probiotic that improves survival under these conditions would reduce mortality and improve productivity.

2. In section 4.1, the methods used for isolation of bacteria are summarised but nothing is mentioned regarding other groups like fungi. While these are less studied, they may be important components of the microbiota that can be beneficial for insect performance and fitness.

3. In the Introduction (section 1) it is stated that prebiotics will be discussed in brief. Very little is actually said on this topic until the concluding remarks. It would be worth adding a short section on prebiotics, even if it is to highlight how little is known, and what is needed to develop them. The knowledge gap can then be reinforced in the concluding remarks as they are currently written.

4. Throughout the review, the term 'fitness performance' is used. I suggest rewriting as 'fitness and performance' because fitness relates to the integration of survival and reproduction, whereas performance is a measure of completion of a task like growth or development.

The review is written well for the most part. There are some spelling, grammatical and sentence structure issues that I have highlighted below and suggested improvements.

Line 16: 'insect diseases' not 'insects diseases'

Line 19: 'cases' not 'case'

Line 35: Replace 'mass-rearing insects confront' with 'mass-rearing of insects faces'

Line 54: Delete 'the'

Line 56-57: Replace 'can distort the sense of smell of the insects changing its feeding behaviour' with 'can distort the feeding behaviour of insects by changing their sense of smell'

Line 59: 'Insect-microorganism' not 'Insect-microorganisms'

Line 95: Replace 'is' with 'are'

Line 125: Replace 'insects’ traits' with 'insect traits'

Line 144: Replace 'microbiota is' with 'microbiota are'

Lines 152-155: This sentence is incomplete. It does not have a verb or object.

Line 192: Replace 'come to play' with 'come into play'

Line 194: trypanosome should not be italicised - it is not a genus or species name. Trypanosoma is the genus name, and should be italicised if appropriate here.

Line 220: Shouldn't Lactobacillus be italicised if it is a genus name?

Line 222: Place a comma before 'respectively'

Line 226: Replace with 'small molecules RNA' with  'small-molecule RNA'

Line 264: Delete 'also'

Table 1. Rewrite the table title as: 

A summary of probiotics tested on insects mass reared for food and feed with the objective of improving insect performance or fitness against natural pathogens in the mass rearing environment. The table does not include data where bacteria/yeast have been provided to the insect as a probiotic to test its efficacy against a specific human pathogen in vivo, nor does it include insects reared for sterile insect technique programmes.

Line 281: Capital 'P' for 'Probiotic'

Lines 315-317. The clause after the comma should be written as a new sentence. It should also include the words 'needs to be quantified' to make it a complete sentence.

Lines 321-325: This very long sentence should be cut into smaller clauses. At the moment it is difficult to understand because there are commas missing that would help to structure it.

Line 325: Replace 'are such as' with 'include'

Lines 339-342: This long sentence does not make sense. There seems to be something missing, but also cut it into separate clauses to improve communication.

Line 346: 'pathogen' not 'pathogens'

Line 350: Replace 'on pathogen’s capacity' with 'the capacity of pathogens'

Line 384: 'diet' not 'Diet'

Line 395 and 401: Replace 'medfly' with 'C. capitata'. The species has been defined in the review before but the shortened common name has not.

Figure 2: I found that the version in the pdf that I had to review was of very poor quality and the text was unreadable. Please check that the resolution is sufficient for it to be viewed and interpreted.

Lines 430-434: This is a very long sentence that loses its way. Cut into smaller sentences. 'The' is also used too often in the sentence and can be removed in some cases.

Line 435: 'fitness and performance' rather than 'fitness performance'

Line 436: 'insect species' not 'insects species'

Lines 435-438: Long sentence. Break into two parts: the first introducing that some evidence has been derived, the second using B. mori as an example.

Author Response

Thank you for the comments and the suggestions, 

please find the corrections and the additions in the attached file. 

Best regards 

Reviewer 2 Report

Manuscript insects-1659941 entitled “Bugs in Bugs: The Role of Probiotics and Prebiotics in Maintenance of Health in Mass Reared Insects” is a well written review and flows well from start to end. It is well suited for the special issue "Insect–Pathogen Interactions in Mass-Reared Insects". I command the authors for the excellent overview of the use of probiotics to the researchers and students of insect mass rearing and recommend consideration for publication as written. Some annotations in the attached PDF for authors’ revision. 

Author Response

Thank you for the comments and suggestions, please find in the attachments all the corrections. 

Best regards 
